# Can athletes be tough *yet* compassionate to themselves? Practical implications for NCAA mental health best practice no. 4

Andreas Stamatis[1]*, Paul J. Deal[2], Grant B. Morgan[3], Jeffrey S. Forsse[4,5], Zacharias Papadakis[6], Sarah McKinley-Barnard[7], Eric M. Scudamore[8], Panagiotis Koutakis[9]

1 Sport and Wellness, SUNY Plattsburgh, Plattsburgh, New York, United States of America, 2 Counselor Education, SUNY Plattsburgh, Plattsburgh, New York, United States of America, 3 Educational Psychology, Baylor University, Waco, Texas, United States of America, 4 Kinesiology and Health Science, Stephen F. Austin State University, Nacogdoches, Texas, United States of America, 5 Health, Human Performance and Recreation, Baylor University, Waco, Texas, United States of America, 6 Sport and Exercise Sciences, Barry University, Miami Shores, Florida, United States of America, 7 Health, Kinesiology, and Sport, University of South Alabama, Mobile, Alabama, United States of America, 8 Health, Physical Education, and Sport Sciences, Arkansas State University, Jonesboro, Arkansas, United States of America, 9 Nutrition, Food and Exercise Sciences, Florida State University, Tallahassee, Florida, United States of America

* coach_stam@rocketmail.com

**Data Availability Statement:** All relevant data are available within the Figshare repository at https://doi.org/10.6084/m9.figshare.13383059.v1.

## Abstract

Recent tragic events and data from official NCAA reports suggest student-athletes' well-being is compromised by symptoms of mental health (MH) disorders. Self-compassion (SC) and mental toughness (MT) are two psychological constructs that have been shown effective against stressors associated with sports. The purpose of this study was to investigate SC, MT, and MH in a NCAA environment for the first time and provide practical suggestions for MH best practice No.4. In total, 542 student-athletes participated across Divisions ($M_{age}$ = 19.84, $SD$ = 1.7). Data were collected through Mental Toughness Index, Self-Compassion Scale, and Mental Health Continuum–Short Form. MT, SC (including mindfulness), and MH were positively correlated. Males scored higher than females on all three scales. No differences were found between divisions. SC partially mediated the MT-MH relationship, but moderation was not significant. Working towards NCAA MH best practice should include training athletes in both MT and SC skills (via mindfulness).

## Introduction

In January 2018, 21-year old Pac-12 football player, Tyler Hilinski, took his own life in his apartment [1]. A year before that, in January 2017, 19-year-old Jordan Hankins, a Big Ten basketball player, died by suicide in her room [2]. Tragic events like these and data from official reports [3] suggest student-athletes' well-being is compromised by sub-clinical and clinical symptoms of mental health (MH) disorders, such as depression and anxiety. According to the World Health Organization, mental wellness is defined as "a state of well-being in which the individual realizes his or her own abilities, can cope with the normal stresses of life, can work productively and fruitfully, and is able to make a contribution to his or her community" [4].

**Funding:** The author(s) received no specific funding for this work.

**Competing interests:** The authors have declared that no competing interests exist.

## NCAA and mental health best practice

Through integrative models that involve more than just physiological processes (e.g., affective) in reaching and sustaining optimal performance (e.g., IMAP) [5], MH has been officially recognized from the field of Sport Psychology as a major component of athletic performance [6]. However, the recognition of this interaction is noteworthy, not only because that was not always the case [7], but also because it shifts the focus of all stakeholders towards promoting athletes' success (in sport *and* life) via a more holistic view.

In the U.S., the National Collegiate Athletic Association (NCAA) started considering putting psychological services to student-athletes into effect almost two decades ago [8]. In 2016, they issued the *Mental Health Best Practice* document in an effort to guide its members in the general prevention/early intervention of MH issues and promotion of MH. Recently, other major American sport organizations, too, turned their attention towards MH. For example, NFL partnered with mentalhealth.gov [9] and NBA with Mental Health America [10]. Therefore, there seems to be a generalized awareness of MH in sports in the US. This study, triggered by the recent catastrophic events presented above, focuses on MH best practices in the NCAA.

NCAA's significant first step of outlining best practices provides some general goals. Through four practices, it mainly address the following key issues: (1) Mental Health care should be provided to student-athletes by licenced and qualified professionals only; (2) Inter-collaborations are encouraged for developing plans that focus on early identification and referral of student-athletes in need; (3) Pre-participation screening for a variety of sub-clinical symptoms of MH disorders, such as depression, anxiety, insomnia, and alcohol use is suggested; and (4) Education of all stakeholders towards the development of sport cultures that promote student-athletes' MH issues management is recommended. This study focuses mainly on the NCAA MH best practice No. 4.

Although that document makes the organization's position clear to all stakeholders, it does not tell us *how* to get there. Recommending ". . .self-care, stress management, and personal health promoting practices" (p. 13) may be essential to best practice No. 4s aspiration to: ". . .support mental well-being and resilience," [3, 11], but what are the specific steps and practices? Like winning seasons, winning MH best practices depend on identifying and implementing the building blocks necessary to achieve long-term success.

Investigating *mental toughness* (MT) and *self-compassion* (SC) as they relate to MH may provide concrete and practical suggestions about specific skills that support best practices. But why focus on MT and SC in particular? As is elaborated below, though an abundance of studies has investigated the positive mental health benefits of these two constructs separately, few studies have considered them in tandem. Qualitative research supports a collaborative relationship between MT [12], but quantitative studies are still needed for corroboration. This study helps clarify, philosophically speaking, if there is justification for viewing these two constructs as complementary, rather than competing approaches to wellness. If athletes being both tough and kind to themselves is supportive of MH, our approach to student-athlete wellness may be philosophically reframed from one of *either/or* to *both/and*. Empirical support for MT and SC corroborates the need for complementary approaches to supporting MH in student-athletes.

NCAA is the largest association of organized collegiate sports in the US with approximately 460,000 student-athletes participating in 24 sports around the year [13]. NCAA is grouped in three Divisions. Division I programs attract the most talented athletes, but also the most attention from all stakeholders (e.g., media, sport agents, fans). Therefore, *ceteris paribus*, athletes in Division I programs likely endure the most stress and scrutiny of the three Divisions [14]. However, according to Morgan's mental health model (the Iceberg Profile) [15], athletes of

higher levels of performance (e.g., NCAA Division I programs) exhibit greater positive mental health than those of lower levels (e.g., NCAA Division III programs).

Regardless of Division, student-athletes' dual-role equates with more stressors, which are further amplified by *stigma* regarding MH problems [16], thereby creating a need for additional support [17]. As a result, coping mechanisms become crucial for healthy adjustment to the stressors of sport, optimal performance, and overall well-being. Self-compassion (SC) and mental toughness (MT) are two psychological constructs that have been initially proven effective against stressors associated with sports.

## Self-compassion

Self-compassion, although its definition in the sporting environment has not reached consensus yet, is conceptualized as containing three core components: *self-kindness versus self-judgment* (extending kindness and understanding to oneself rather than harsh judgment and self-criticism); *common humanity versus isolation* (the ability to see difficult experiences as a part of a larger human experience rather than seeing them as isolating personal failures); and *mindfulness versus over-identification* (holding and noticing one's thoughts and emotions in balanced awareness without identifying with them) [18]. While many constructs link to MH, research on the key factors of therapeutic action, suggest SC is essential [19]. In the case of reducing depressive symptoms, an experimental study [20] showed SC was more effective at regulating negative emotions than cognitive appraisal or acceptance conditions.

While not very different from similar constructs, such as compassion [21] or body-self compassion [22], SC teaches the athlete to relate to painful experiences in sport, such as difficulties and failures, without resorting to excessive self-criticism, isolation, and over-identification [23].

This useful, but under-utilized, psychological process [24] has been found valuable when athletes need to reduce negative sport-triggered emotions [24, 25], to promote constructive criticism during difficult times [25, 26], to avoid rumination over faults [26, 27], to foster well-being when exercising [28], and to substitute *global self-esteem* with a healthier alternative [29]. It is noteworthy that males have been found to score higher in SC scales than females [30].

Concerning stigma, SC has been seen as a way for vulnerability to no longer be ignored in sport psychology research and practice [31]. In fact, SC has been suggested as the "upside of vulnerability" that can support and maintain MH in high-demand sport settings [32, p.6]. Theoretically, *self-kindness* ameliorates vulnerable feelings of seeing oneself as inherently flawed, *common humanity* brings individuals out of the vulnerable position of isolation and back into community, and *mindfulness* meets instances of vulnerable exposure with equanimity. In all three cases, SC provides tools for working effectively with the vulnerability of performing as a student-athlete. Overall, with the possible exception of one study in which women athletes perceived that SC might lead to mediocrity [33], findings suggest that SC can be beneficial to athletes.

## Mental toughness

More than 60 years ago, Cattell et al. [34] made one of the pioneering attempts to explain the term (i.e., *tough-mindedness*). Three decades later, the concept was updated and promoted by Loehr [35]. Since then, MT has been increasingly associated with successful performance in several stressful and competitive environments (e.g., the military, business, academics, medicine), including sports [36].

Nevertheless, despite the strong interest MT has received in research and practice [37], a plethora of MT definitions and several conceptualization frameworks frustrate concensus [38].

For example, Gucciardi [39] conceptualizes MT as uni-dimensional, where psychological dimensions assimilate over time due to commonalities [40]. Others, such as Crust [41], support MT as a multi-dimensional construct—based on the integrative personality framework–comprised of multiple layers of personality [42].

Recently, a shift towards less variation has been identified [43]. Uni-dimensionality and Gucciardi's definition (i.e., caravan of personal resources) have started gaining acceptance in the scientific community [44]: "A personal capacity to produce consistently high levels of subjective (e.g., personal goals or strivings) or objective performance (e.g., sales, race time, GPA) despite everyday challenges and stressors as well as significant adversities" (p.28) [45]. For more information concerning definition (what MT is and what it is not), conceptualization (uni- vs. multi-dimensional models) and measurement of MT (global vs. specific view on inventories) [46].

Concerning MH, although Bauman [47] suggested that MT and MH are contradictory concepts on the elite level due to sport culture, Gucciardi et al. [48] view that conclusion as premature. MT has been correlated not only with predictors of performance, but also with positive MH outcomes. For example, MT has been associated positively with motivation [49, 50] and self-efficacy [51]. MT has also been interrelated negatively with stress [52] and depression [53], and positively with thriving [54], sleep quality [55], psychological well-being [56], and MH [57]. Of note, preliminary evidence suggests that males present higher levels of MT than women [51].

Concerning stigma, it has been proposed that athletes could perceive "MT training" as a more attractive alternative than traditional MH services [48]. However, coaches need training in order to position themselves as facilitative of environments that support MH [50, 58].

Overall, findings suggest that MT can be beneficial to athletes. Nonetheless, there may be a negative aspect of MT. In particular, high levels of MT have been associated with negative behaviors during rehabilitation [59], such as adherence to recovery protocols.

## Mental toughness and self-compassion

The conceptualizations of MT and SC in sport seem contradictory. MT is hypothesized as a key attribute in constant performance and consistent thriving despite stress levels and it is usually portrayed through a "machismo" mentality [48, 60–62]. Conversely, SC's operationalization entails self-kindness and arresting self-objectification and social comparison when facing setbacks [63]. Can toughness and kindness work together to support MH?

Despite these differences, MT and SC also share a lot of commonalities. In the context of sport culture, the definitions of both still seem to lack clear consensus, both have been positively correlated with MH outcomes, both have shown promise towards eliminating MH stigma, both have a negative aspect, and both present significant differences based on gender; Concerning the latter, gender significantly affects MH scores, too. For more information, see [64].

These two constructs, which are related but distinct, have been under-investigated as part of a common study. To the best of our knowledge Wilson et al. [12] were the first to explore the compatibility of the two constructs towards athletic performance. Based on qualitative interviews of seven elite women athletes at two separate times, they concluded that: (a) the connection between SC and MT is underpinned by *mindfulness*, (b) SC is required to use MT effectively, and (c) MT and SC are compatible processes (the "zipper effect"). However, MT and SC have never been investigated together through the lens of MH in sporting environments.

## Purpose of study

Based on the information presented above, the purpose of the study was to investigate SC, MT, and MH in a NCAA environment so as to provide evidence towards updating current MH best practices.

The specific objectives included:

a. To confirm (or add on the current, preliminary literature evidence):

- The positive relationships between SC and MH, MT and MH, and SC and MT;

- Differences in scores based on gender;

- Compatibility of MT and SC; and

- Mindfulness underpinning the MT-SC relationship and

b. To explore:

- Possible significant differences of scores based on division and

- The interaction effect of MT-SC on MH.

## Hypotheses

1. MT and MH, SC and MH, and SC and MT are positively correlated.

2. Males score higher in MT, SC, and MH scales.

3. Division I athletes score higher in MT, SC, and MH.

4. The interaction effect of MT and SC on MH is buffering (i.e., the moderator weakens the effect of the predictor on the outcome.).

5. SC will mediate the relationship between MT and MH.

6. MT and SC are compatible via mindfulness.

## Materials and methods

The Ethics Committee of the State University of New York at Plattsburgh approved this study (Number: 1564). Consent Form was obtained electronically through Qualtrics. Below, the authors provide details that could assist with replication of the study. This section consists of information about the sample, the instruments chosen to collect data, the recruitment and data collection procedures, and the statistical analyses used.

### Participants

In total, 542 NCAA student-athletes from all three NCAA Divisions (convenience samples) agreed to participate ($M$age = 19.84, $SD$ = 1.7). The sample was predominantly White, female, Division I (DI) student-athletes. Please, see Table 1 for demographics.

### Instruments

Three separate instruments were used to collect data concerning the three different constructs. The Mental Toughness Index (MTI) was used to collect MT scores, the Self-Compassion Scale (SCS) for SC scores, and the Mental Health Continuum–Short Form (MHC-SF) for MH

**Table 1. Demographic information about participants ($n$ = 542).**

| Variable | Count | Percentage |
|---|---|---|
| Sex | | |
| Male | 227 | 41.9 |
| Female | 315 | 58.1 |
| Race | | |
| Asian/Pacific Islander | 9 | 1.7 |
| Black/African-American | 69 | 12.7 |
| Hispanic/Latino | 70 | 12.9 |
| White/Caucasian | 378 | 69.7 |
| Other | 16 | 3.0 |
| Division | | |
| DI | 229 | 42.26 |
| DII | 122 | 22.50 |
| DIII | 191 | 35.24 |

scores. All three inventories are self-report and were administered to all participants in the following random order: MTI, SCS, and MHC-SF (order effects were not ruled out).

**Mental toughness index.** MTI was developed by Gucciardi, Hanton (45) based on the uni-dimensional conceptualization of the construct and the definition presented above. It includes eight items and the scores can range from 1 (False, 100% of the time) to 7 (True, 100% of the time). Item 1 measures *generalized self-efficacy*, item 2 *emotion regulation*, item 3 *attention regulation*, item 4 *success mindset*, item 5 *buoyancy*, item 6 *overcoming adversity*, item 7 *context knowledge*, and item 8 *optimistic style*.

The authors chose MTI because of the amount and quality of the evidence supporting its reliability and validity [45, 65–67]. MTI was also developed for MT in sport. The estimated reliability estimates, $\alpha$, was .90 in the overall sample.

**Self-compassion scale.** SCS was developed by Neff [18] and consists of 26 items. SCS was conceptualized on the basis of six sub-scales: self-kindness, common humanity, mindfulness, self-judgment, isolation, and over-identification. When calculating the mean of the sub-scales, the last three require reverse coding. Item scores range from 1 (almost never) to 5 (almost always). The self-kindness and self-judgment sub-scales include four items; all other sub-scales are represented by five items each.

The authors chose SCS because it has sound reliability and validity evidence [18, 68, 69] and has been used before with success in the sporting context [27, 28, 70]. The estimated reliability estimates, $\alpha$, was .90 in the overall sample.

**Mental health continuum–short form.** MHC-SF is a 14-item, six-point scale, which was developed by Keyes [71]. The responses indicate how participants felt over the past month. It contains three dimensions of wellbeing: (a) emotional (items 1–3), (b) social (items 4–8), and (c) psychological (items 9–14). MHC-SF assesses MH along a continuum of languishing to flourishing. MH is operationalized as positive MH.

The authors chose MHC-SF because it has been successfully evaluated psychometrically [72–74] and it has been used in the sporting context in the past [50, 75]. The estimated reliability estimates, $\alpha$, was .93 in the overall sample.

## Procedure

After the Ethics Committee approved the study proposal, permission was given for data collection almost anywhere in the US under the specific conditions described in the project. The

three questionnaires were uploaded on Qualtrics and were distributed to three DI, one DII, and one DIII institutions competing in the conferences of Big12, Sunshine State, South Belt (two), and SUNYAC. The student-athletes from those institutions were invited to participate via email. The emails included individualized links. The consent form was obtained electronically. The Mobile Friendly option was activated. Recruitment stopped after three reminder emails.

## Statistical analyses

The statistical analyses used in this study were determined by each of the research hypotheses. For Hypothesis 1, the authors estimated all bivariate correlations and generated a scatterplot matrix to visually examine the strength, direction, and linearity of the relationships. For Hypothesis 2, the authors simultaneously tested the equality of the three mean vectors–MT, SC, and MH–using Hotelling's $T^2$ test. For Hypothesis 3, the authors conducted three one-way analyses of variance (ANOVA) to examine whether there were differences between divisions on the three outcomes. Each of the analyses for Hypothesis 1–3 used a maximum allowable Type I error rate of 5%. For Hypothesis 4, the authors estimated a structural equation model (SEM) with the MT and SC latent variables as well as their latent interaction (i.e., moderation effect) as explanatory variables of MH. The model was estimated using maximum likelihood with robust standard errors. For Hypothesis 5, the authors examined the mediation effect of SC on the relationship between MT and MH using SEM. The mediation effect was estimated via the indirect of effect of MT on MH by way of SC. For Hypothesis 6, the authors examined the latent correlation between MT and the Mindfulness latent variables; Mindfulness is one of the constructs that makes up SC. Analyses for testing Hypotheses 1–3 were completed in IBM SPSS Statistics [76], and those for testing Hypothesis 4–6 were completed in Mplus (version 8.2) [77].

## Results

First, descriptive statistics were generated for MT, SC and MH scores overall and by gender and division (see Tables 2 and 3, below). Then, the authors conducted the necessary inferential procedures.

### Hypothesis 1: MT and MH, SC and MH, and SC and MT are positively correlated

Evidence for Hypothesis 1 was found. In more detail, the bivariate correlations were indeed all positive and moderately strong. The correlation between MT and SC scores was .44, MT and MH scores was .41, and SC and MH scores was .54. The scatterplot matrix is shown in Fig 1. Clearly, there was a ceiling effect for the MT and MH scores.

### Hypothesis 2: Males score higher in MT, SC, and MH scales

Evidence for Hypothesis 2 was found. In particular, the results of the Hotelling's $T^2$ test indicated that there was a difference between the mean vectors of the three variables on the basis of gender. This finding indicates that there were mean differences on at least one of the three variables ($F[3, 538] = 14.78$, $p < .001$). About 8% of the variability in the mean vectors was attributable to gender, which is a small to medium effect. To determine where the differences were, the authors conducted three post-hoc independent samples $t$ tests using a Type I error rate of 5%. Given the directionality of the hypothesis (i.e., males score *higher* than females), the tests all examined the one-sided p-value. Based on the post-hoc $t$ tests, males did score higher on

**Table 2. Descriptive statistics of all scales and subscales by gender.**

| Scale or Subscale | Male | | Female | | Overall | |
|---|---|---|---|---|---|---|
| | **M** | **SD** | **M** | **SD** | **M** | **SD** |
| Mental Toughness | 47.5 | 6.8 | 44.5 | 6.3 | 45.7 | 6.7 |
| Mental Health | 51.5 | 12.5 | 48.9 | 12.3 | 50.0 | 12.4 |
| Self-Compassion | 3.3 | 0.5 | 3.0 | 0.6 | 3.2 | 0.6 |
| Self-Knowledge | 3.2 | 0.8 | 3.0 | 0.7 | 3.1 | 0.8 |
| Self-Judgment | 3.1 | 0.8 | 2.7 | 0.8 | 2.9 | 0.8 |
| Common Humanity | 3.2 | 0.8 | 3.2 | 0.8 | 3.2 | 0.8 |
| Isolation | 3.4 | 0.9 | 3.0 | 0.9 | 3.2 | 0.9 |
| Mindfulness | 3.5 | 0.8 | 3.3 | 0.7 | 3.4 | 0.8 |
| Over-identification | 3.5 | 0.9 | 3.0 | 0.8 | 3.2 | 0.9 |

Note. Mental Toughness (MT) scores are the sum of responses across eight items; higher scores are associated with more MT. Mental Health (MH) scores are the sum of responses across 14 items; higher scores are associated with better MH. Self-compassion (SC) scores are the mean responses across 26 items; high scores are reflective of better SC. Self-knowledge, self-judgment, common humanity, isolation, mindfulness, and over-identification are subscales of the SC scale. Subscale scores are the means of responses to a subset of items on the self-care scale. Higher scores reflect positive outcomes because the isolation, self-judgment and over-identification are reverse scored.

average than females on MT ($M_M$ = 47.53, $M_F$ = 44.45; $t$ = 5.45, $p < .001$, d = 0.47), SC ($M_M$ = 3.32, $M_F$ = 3.04; $t$ = 2.43, $p = .008$, d = 0.49), and MH ($M_M$ = 51.52, $M_F$ = 48.90; 5.47, $p < .001$, $d = 0.21$). The standardized mean differences for MT and SC were moderately large, and the standardized mean difference in MH scores was small.

## Hypothesis 3: Division I athletes score higher in MT, SC, and MH

Evidence for Hypothesis 3 was not found. There were no differences between divisions on MT ($F$ = 1.56, $p = .21$, $\eta_P^2 = .006$), SC ($F$ = 2.12, $p = .12$, $\eta_P^2 = .008$), or MH ($F$ = 1.01, $p = .36$, $\eta_P^2 = .004$).

**Table 3. Descriptive statistics of all scales and subscales by NCAA division.**

| Scale or Subscale | Division I | | Division II | | Division III | | Overall | |
|---|---|---|---|---|---|---|---|---|
| | **M** | **SD** | **M** | **SD** | **M** | **SD** | **M** | **SD** |
| Mental Toughness | 45.5 | 6.7 | 46.7 | 6.3 | 45.4 | 6.8 | 45.7 | 6.7 |
| Mental Health | 49.6 | 12.8 | 48.6 | 12.1 | 51.4 | 12.2 | 50.0 | 12.4 |
| Self-Compassion | 3.1 | 0.6 | 3.2 | 0.5 | 3.2 | 0.6 | 3.2 | 0.6 |
| Self-Knowledge | 3.1 | 0.8 | 3.1 | 0.7 | 3.1 | 0.7 | 3.1 | 0.8 |
| Self-Judgment | 2.8 | 0.8 | 2.9 | 0.8 | 3.0 | 0.8 | 2.9 | 0.8 |
| Common Humanity | 3.2 | 0.8 | 3.2 | 0.8 | 3.2 | 0.8 | 3.2 | 0.8 |
| Isolation | 3.1 | 1.0 | 3.2 | 0.9 | 3.2 | 0.9 | 3.2 | 0.9 |
| Mindfulness | 3.4 | 0.7 | 3.4 | 0.8 | 3.4 | 0.7 | 3.4 | 0.8 |
| Over-identification | 3.2 | 0.9 | 3.2 | 0.8 | 3.3 | 0.9 | 3.2 | 0.9 |

Note. Mental Toughness (MT) scores are the sum of responses across eight items; higher scores are associated with more MT. Mental Health (MH) scores are the sum of responses across 14 items; higher scores are associated with better MH. Self-compassion (SC) scores are the mean responses across 26 items; high scores are reflective of better SC. Self-knowledge, self-judgment, common humanity, isolation, mindfulness, and over-identification are subscales of the SC scale. Subscale scores are the means of responses to a subset of items on the self-care scale. Higher scores reflect positive outcomes because the isolation, self-judgment and over-identification are reverse scored.

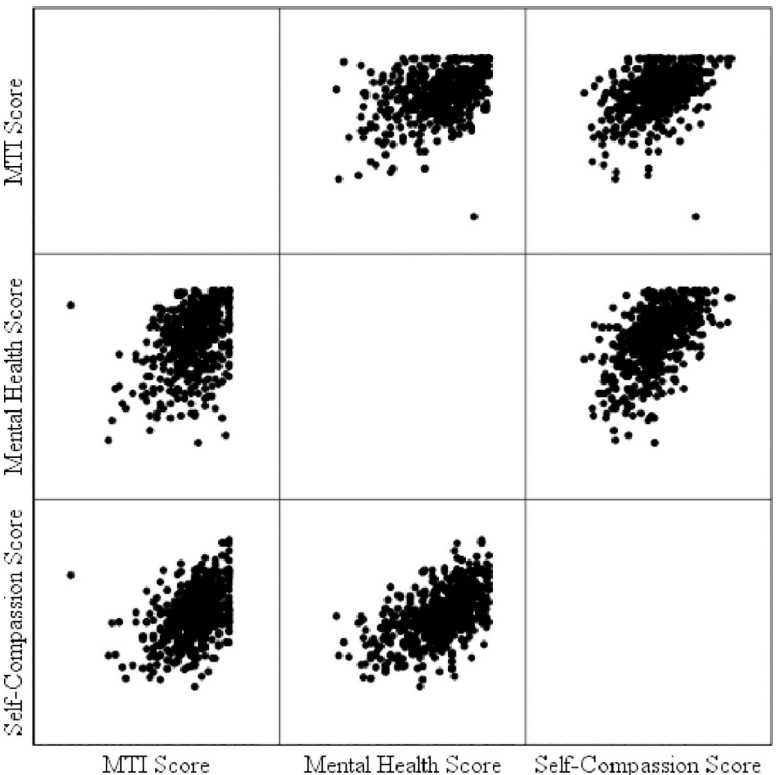

**Fig 1. Scatterplot matrix of MT, SC, and MH scores.**

### Hypothesis 4: The interaction effect of MT in the SC-MH relationship is buffering

Hypothesis 4 was supported descriptively, but could not be supported inferentially. Overall, the model explained about 41.6% ($R^2$, Z = 10.38, $p < .001$) of the variability in MH. There were statistically significant main effects for MT ($\gamma_{MT}$ = 0.23, Z = 4.43, $p < .001$) and SC ($\gamma_{SC}$ = 0.50, Z = 10.48, $p < .001$); MT and SC were also significantly correlated ($\varphi$ = .49, Z = 9.76, $p < .001$). As for the latent interaction between MT and SC, there was a negative, but non-significant moderation effect ($\gamma_{MT \times SC}$ = -0.06, Z = -1.44, $p = .15$). To compare the relative improvement in fit using information criteria, the authors examined the Akaike information criterion (AIC) and Bayesian information criterion (BIC) between the structural models with and without the latent interaction between MT and SC. Both AIC and BIC are both relative measures of model fit where smaller values indicate better fit between competing models. The AIC estimates for the model with and without the latent interaction were 61229.3 and 61230.2, respectively, whereas the BIC estimates for the same models were 61316.0 and 62312.7, respectively. These estimates indicate that the model that includes the moderation effect is a better fitting model, but there are no guidelines for how much lower an estimate needs to be in order for the improvement in fit to be "meaningful." The full structural model results are shown in Fig 2.

### Hypothesis 5: SC will partially mediate the relationship between MT and MH

Hypothesis 5 was supported in that SC partially mediated the relationship between MT and MH. Using SEM, the authors estimated the direct effect of MT on MH as well as the indirect

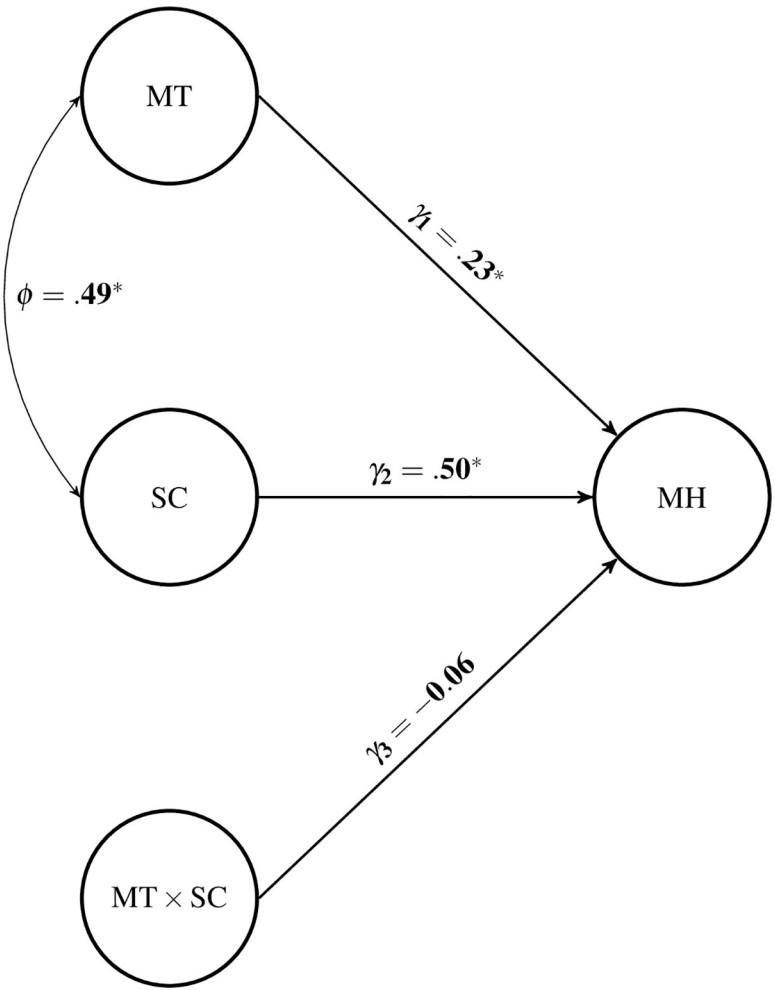

**Fig 2. Structural parameter estimates from moderation model.**

effect of MT on MH through SC. The indirect effect is the partial mediation of SC, which can be viewed as the effect MT has on MH when SC is present and activated. The direct effect estimate of MT on MH was .23 (Z = 5.28, p < .001), and the indirect effect through SC was .22 (Z = 8.24, p < .001). The total effect of MT on MH is the sum of the direct and indirect effects, .45. The mediation model explained 19.3% of the variability in SC with MT, and 38.9% of the variability in MH with MT and the partial mediation of SC. The structural estimates are presented in Fig 3; that is, the manifest variables and their loadings and residual variances were excluded from the diagram for clarity.

### Hypothesis 6: MT and SC are compatible via mindfulness

Hypothesis 6 was supported using the output from the latent variable model in which the subscales of SC were allowed to correlate with each other as well as with MT. Each of the SC subscales was correlated with MT, but the correlation between MT and the Mindfulness construct was strongest ($\varphi$ = .45, Z = 12.15, p < .001). The standardized factor correlations with MT are shown in Table 4.

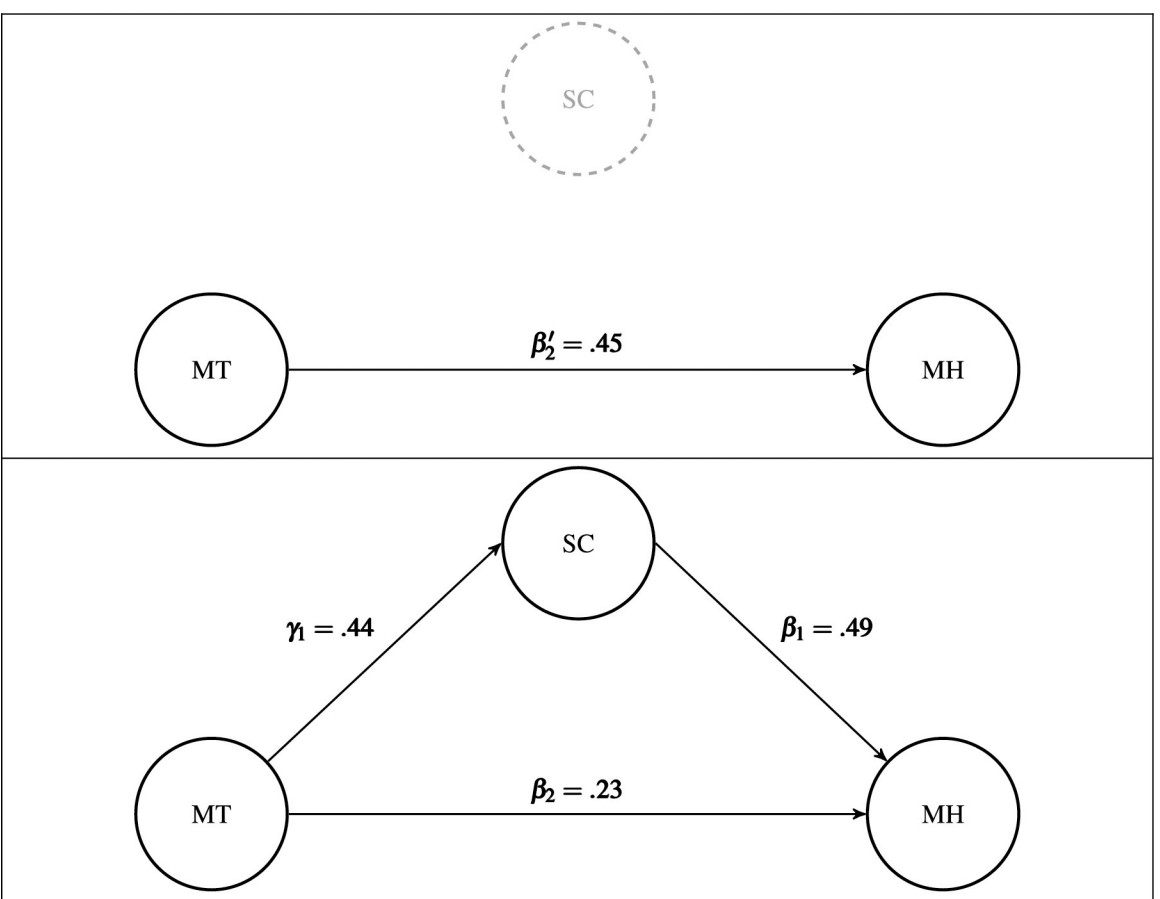

**Fig 3. Structural mediation model results.** The mediation (i.e., indirect) effect of SC is equal to the difference between $\beta'_2 - \beta_2 = .45 - .23 = .22$, which is also equal to the products of the paths with SC ($\gamma_1 \times \beta_1 = .44 \times .49 = .22$).

## Discussion

The findings coalesce around a common theme—MT and SC are distinct, but related, constructs supportive of positive MH in student-athletes across gender and NCAA Division. As such, they offer another piece of support to understanding the role of SC in sport [78]. Moreover, they offer the novel contribution of bringing together two parallel constructs—MT and SC—investigated with regard to MH in sport. Above, we drew attention to the need for greater clarity and direction about how and what promotes MH best practices for collegiate athletes. Findings offer grounds to suggest that training in both MT and SC skills (via *mindfulness)* may

**Table 4. Standardized factor correlations for SC subscales and MT.**

|      | SJ  | SK  | CH  | I   | M   | MT  |
|------|-----|-----|-----|-----|-----|-----|
| SJ   | –   |     |     |     |     |     |
| SK   | .55 | –   |     |     |     |     |
| CH   | .23 | .66 | –   |     |     |     |
| I    | .87 | .44 | .18 | –   |     |     |
| M    | .29 | .80 | .73 | .36 | –   |     |
| MT   | .35 | .42 | .33 | .38 | .45 | –   |

Note. The OI subscale was excluded from this analysis due to convergence problems.

be one specific step towards achieving that end. In the following, each hypothesis is briefly reviewed and the suggestion is discussed as it relates to next practical steps towards realizing NCAA mental health best practice No. 4.

## Review of six hypotheses

Building on previous research, our support for Hypothesis 1 (positive correlations between MT and MH, SC and MH, SC and MT), is in agreement with the literature [12, 48, 49]. MT and SC correlate positively with MH. In other words, the main effects indicate both can support MH. Therefore, training athletes in both MT and SC skills is a tangible step towards promoting mental health in athletic culture (See Practical Implication below).

In support of the second hypothesis, males scored higher in MT, SC, and MH scales [30, 51, 64]. While all athletes across the gender spectrum can benefit from training in MT and SC skills, women's lower scores indicate a need for further exploration. Sexism is detrimental to MH, but it is difficult to determine the meaning of women's lower scores. Sport culture, in part, has absorbed some of culture's sexist attitudes [79–81]. Female athletes may have internalized a sense of being "less than" or inferior in comparison to the dominant norm of male sport. Having to fight for equal pay in women's sport is one such example. The role of cultural gender norms must be considered a factor in how all athletes across the gender spectrum relate to constructs of toughness and kindness (See Recommendations for Future Studies below).

No support was found for the third hypothesis, which is that Division I athletes score higher in MT, SC, and MH. Consequently, our results indicated that no special MT, SC, and MH psychological skill training based on Division should be suggested.

Evidence was not found for the fourth hypothesis. In more detail, although the interaction effect of MT in the SC-MH relationship is buffering in this sample, it is not statistically significant. So, chance observation of the buffering relationship cannot be ruled out. These results warrant further elaboration.

Concerning the fifth hypothesis, evidence was found. The results of the mediation analysis endorse what Wilson et al. [12] suggest, which is that SC is instrumental to facilitation of MT via statistically-significant direct and indirect effects (i.e., partial mediation). In other words, SC may support student-athletes with facing hardship. Wilson et al. [12], suggested that—if one cannot move forward after facing adversity—they may not be able to sustain a mentally-tough attitude. This is another piece of evidence supporting the compatibility of MT and SC (See Practical Implication below).

Evidence for the sixth and last hypothesis was found. A key connector between MT and SC is *mindfulness*. Mindfulness was fundamental to increasing and conserving both SC and MT (This evidence is again used in Practical Implication below). The women participants of Wilson et al. [12] used it for *attention-* and *emotion-regulation* (both key dimensions of Gucciardi's conceptualization of MT). This may also support the notion that *mindfulness* should be further investigated as a possible component of, not only SC, but MT, as well (e.g., possible additional *key dimension* in Gucciardi's unidimensional conceptualization of MT?).

While many constructs link to MH, research on the key factors of therapeutic action, suggest SC is essential [19]. In the case of reducing depressive symptoms, an experimental study [20] showed SC was more effective at regulating negative emotions than cognitive appraisal or acceptance conditions. The authors conclude with limitations and recommendations for future studies.

## Practical implication: Cultivate skills in both MT and SC (via mindfulness)

Despite a statistically non-significant buffering interaction effect, the authors showed evidence to support the compatibility of MT and SC. Consequently, although more research is still

needed to clarify the combined effect of MT and SC on MH, our findings (i.e., SC and MT are positively correlated, SC is crucial for using MT efficiently, there is common ground between MT and SC and it is called *mindfulness*), may support a provisional suggestion that cultivating skills in both MT [82] and SC [27] is a practical step towards promoting wellness in student-athletes. Moreover, it appears that coaches, athletes, and certified mental performance consultants, interested in how to implement NCAA best practice No. 4, will benefit to understand the value of cultivating both through the common factor of *mindfulness*.

Instituting this practical implication involves addressing barriers within sports culture. Coaches and athletes conditioned in the toughness mind-set common to competitive sport culture may understandably balk at the perceived "softness" of SC (self-kind vs. self-critical) [12]. Concerns about losing one's competitive edge must be addressed. One way to do accomplishing this step might involve framing MT and SC as two different, but related, mind-sets for engaging in collegiate sport. Like derivative languages, they both trace to a common origin in mindfulness. Though the world of competitive sport may favour MT as a primary language, athletes may learn to speak SC through intentional effort until it, too, becomes habitual. Intuitive as it may seem to conceptualize MT and SC as two ends of a continuum, this is incorrect. MT is not simply the absence of SC and SC is not simply the absence of MT; they are separate skills with the potential to complement one another in supporting MH. Athletes can be both tough and kind to themselves with benefits to mental health.

## Understanding gender differences: Considering sport culture and gender norms

Promoting MH in NCAA sport culture requires a more nuanced understanding as to why women score lower in MT and SC. Admittedly, because the etiology of those different scores is not entirely understood, making definitive recommendations is still premature. Areas for further research that may clarify gender differences include attention to sport culture and cultural gender norms. As noted above, normative sport culture seems to align more closely with male, rather than female gender norms—competitive stoic toughness is more acceptable than passivity or emotional sensitivity. Additionally, although gender norms are evolving and becoming less rigid (e.g., Michael Phelps disclosure of struggles with depressive symptoms), they create a kind of straitjacket that restrict behaviors beyond the norm. Male athletes may suffer silently with MH problems because gender norms restrict the expression of emotion and mask distress behind a stoic exterior [83]. Female athletes, however, may suffer simply for engaging in sport because female gender norms prescribe that women be sensitive, passive and agreeable, thereby restricting expressions of competitiveness and assertiveness common to sport. They may simultaneously relish the thrill of competition, yet suffer the consequences of violating internalized standards. While gender norms likely harm athletes across the gender continuum, the masculine bias in sport itself might exacerbate MH stress in females.

In terms of the gender differences between SC scores, gender role orientation is an area in need of study for student athletes. Gender itself—checking "male" or "female"—has the power to influence norms of behavior, but as Yarnell et al. [84] argue, gender role orientation captures the extent one identifies with and adheres to socialized "masculine" and "feminine" gender role norms. The effects of gender largely depend on gender role orientation. For instance, though Yarnell et al. [84] found lower SC scores in women than men, they also found that both men and women socialized with a gender role orientation of androgynous and masculine traits had higher levels of SC than men and women socialized to adhere to feminine traits. In short, gender role orientation was a more robust predictor of SC than gender alone. Research should investigate if this finding persists among collegiate athletes. In conclusion and based on the results, the authors

recommend that researchers and stakeholders concentrate on unpacking the differences in SC related to sport culture and socialized gender roles (See Future Recommendations below).

## Conclusions

Investing in athletes is about more than just preventing mental disorders. It is also about promoting MH and wellness. Inviting athletics directors, MH clinicians and practitioners (e.g., sport psychologists, sport psychology consultants), coaches, and athletes to broaden sports training to include MT and SC could support the well-being of collegiate athletes.

Sport provides the occasion to cultivate mental skills conducive to the wellness of the whole person. Various schools of ancient Greek philosophy developed mind-body practices, including reflecting on their vulnerabilities, as a means to perfect and transform oneself [85]. Sport carries this invitation too, particularly if equipped with the skills to approach vulnerability constructively and reframe it as part of growth. It seems that MT and SC help provide these skills.

Good hygiene is preventative. Mental hygiene helps prevent mental distress by caring for one's mind. Mind training is a primary method of mental hygiene that strengthens the neural structures involved with focusing attention and regulating physiological distress [86]. MT and SC represent two such types of mind training. These sorts of psychological tools can equip athletes with skills that may outlast their college experiences. Performance in sport may remain the ultimate goal, but MT and SC are skills that also happen to bolster positive psychological health and wellness in general.

Finally, as noted above, best practices provide general goals, but they do not clarify the specific skills and education needed for their realization. The authors believe their suggestion offers a preliminary step as to *how* to realize the "health-promoting environments that support mental well-being and resilience" put forth in NCAA mental health best practices No. 4 [11]. In particular, the authors have described how psychological tools of MT and SC can act as "self-care, stress management, and personal health promoting practices" (p. 13). Building on these suggestions will involve identifying specific training methods and techniques for teaching MT and SC to student-athletes and coaches. One promising place to start with regard to SC is to adapt and observe the effects of the Mindful Self-Compassion Program, an eight-week, 24-hour training involving guided meditation and discussion sessions [87]. If an equivalent program for MT were developed, that, too, might be applied and evaluated.

Possible limitations of this study include self-assessment and athletes representing five institutions only. The former questions the degree of accuracy of the reported statements. In other words, although self-reported data are essential to the health care field, the personal bias included in these answers raises scepticism [88]. The latter affects the generalizability of the inferences of these results. In more detail, the sample is not representative of the population in terms of gender, race, or Division [13].

One unresolved issue that deserves further examination is the interaction effect of MT and SC towards MH. Is it buffering, enhancing, or antagonistic? For example, in light of the buffering effect observed, attempting to speak the languages of MT and SC simultaneously may detract from their efficacy. Thus, optimal MH in sport may not just be about speaking both languages, but the timing and the context of their application. A skilful use of MT and SC might involve knowing which language is best for a given situation. Optimum use of these skills would require adeptness, but also discernment. For instance, might MT be optimal for moments of action and SC for process and review: MT during competition and SC afterwards? Preliminary research suggests context is a central factor [12], but the non-significant buffering interaction effect of our research indicates that more investigation is needed to understand how and when to exercise these different skills in supporting optimal MH.

Similar, larger-scale research projects are needed in the future. More schools will increase the degree of the extrapolation of the findings since the representation of the sample will be closer to the whole NCAA population. In addition, triangulation, via multi-rating (e.g., parents, coaches, certified mental performance consultants), will increase the validation of the data. Cross-verification though multiple resources, although time-consuming, offers additional benefits, such as investigation of the degree of compatibility of the different assessors recognizing theoretical constructs. Based on these preliminary data, expansion of best practices may be further aided with experimental studies on MT and SC. Moreover, and especially since Yarnell et al. [30, 84] concluded that their results on gender differences on SC should not be overemphasized, future research should continue examining gender differences. In relation to that, and since the current study did not provide an opportunity for participants to identify as a gender other than male or female (delimitation of this study), future designs should try to avoid gender bias, since that may lead to the lack of incorporation of gender data into evidence-based practice. Lastly, although there are no concerns with the instruments adopted by the authors, there are other instruments that could be used for data collection, such as the Sports Mental Toughness Questionnaire (SMTQ) [87], the Self-Compassion Scale-Short Form (SCS-SF) [68], and the MOS 36-item short-form health survey (SF-36) [89].

## Author Contributions

**Conceptualization:** Andreas Stamatis, Paul J. Deal.

**Data curation:** Andreas Stamatis, Paul J. Deal, Grant B. Morgan, Jeffrey S. Forsse, Zacharias Papadakis, Sarah McKinley-Barnard, Eric M. Scudamore.

**Formal analysis:** Grant B. Morgan.

**Funding acquisition:** Panagiotis Koutakis.

**Investigation:** Andreas Stamatis, Grant B. Morgan, Jeffrey S. Forsse, Zacharias Papadakis, Sarah McKinley-Barnard, Eric M. Scudamore.

**Methodology:** Andreas Stamatis, Paul J. Deal, Grant B. Morgan.

**Project administration:** Andreas Stamatis, Jeffrey S. Forsse, Zacharias Papadakis, Sarah McKinley-Barnard, Eric M. Scudamore.

**Resources:** Sarah McKinley-Barnard, Eric M. Scudamore.

**Supervision:** Andreas Stamatis.

**Visualization:** Andreas Stamatis.

**Writing – original draft:** Andreas Stamatis, Paul J. Deal, Grant B. Morgan.

**Writing – review & editing:** Andreas Stamatis, Paul J. Deal, Grant B. Morgan, Zacharias Papadakis, Panagiotis Koutakis.

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
