## [Decision Letter · Decision Letter 0]

15 Sep 2020

PONE-D-20-08466

Can athletes be tough yet kind to themselves? Practical Implications for NCAA Mental Health Best Practice No. 4.

PLOS ONE

Dear Dr. Stamatis,

Thank you for submitting your manuscript to PLOS ONE. After careful consideration, we feel that it has merit but does not fully meet PLOS ONE’s publication criteria as it currently stands. Therefore, we invite you to submit a revised version of the manuscript that addresses the points raised during the review process.

I wanted to raise your attention to an issue we identified during our checks. In particular, your manuscript contains a policy discussion component and we were concerned that the recommendations made in the manuscript and how they are framed (i.e., as recommendations rather than as hypotheses) may not be sufficiently supported by the data presented in the study. The journal does not publish essays, opinion pieces, policy papers, or clinical guidelines, and our publication criteria state that authors may discuss possible implications for their results as long as these are clearly identified as hypotheses instead of conclusions (https://linkprotect.cudasvc.com/url?a=https%3a%2f%2fjournals.plos.org%2fplosone%2fs%2fcriteria-for-publication%23loc-4&c=E,1,osMzIGy4ldYZbUyk4JjVbYWZPw3nlVQDt-5p21CGOvBzbQWbRTJjy-i1V2arhD2QrXQY-JuKmObJtMCkx_Inu_3Ocdk08fMAuNROT0O_fd32Yo0xZivwJLCn&typo=1). The reviewers (reviewer 2 in particular) also highlights this point. Please could you address this in accordance with the PLOS ONE publication criteria to present your conclusions as hypotheses rather than recommendations. Please also address or respond to the reviewers comments below.

We look forward to receiving your revised manuscript.

Kind regards,

Professor Dominic Micklewright, PhD CPsychol PFHEA FBASES FACSM

Academic Editor

PLOS ONE

2. Please consider whether individual events like those listed in the introduction are necessary to your presentation and 1) illustrative of mental health practice issues in the NCAA, and 2) related to the release of the NCAA's 2016 Mental Health Best Practice document since they took place after it.

Reviewers' comments:

Reviewer's Responses to Questions

**Comments to the Author**

1. Is the manuscript technically sound, and do the data support the conclusions?

Reviewer #1: Yes

Reviewer #2: Yes

2. Has the statistical analysis been performed appropriately and rigorously? 

Reviewer #1: Yes

Reviewer #2: Yes

3. Have the authors made all data underlying the findings in their manuscript fully available?

Reviewer #1: No

Reviewer #2: Yes

4. Is the manuscript presented in an intelligible fashion and written in standard English?

Reviewer #1: Yes

Reviewer #2: Yes

5. Review Comments to the Author

Reviewer #1: General Comments

The authors conducted an ambitious study to examine the potential relationship between a measure of Mental Toughness (MT) and Self-Compassion (SC) and the association of these constructs on mental health in a large sample of U.S college varsity athletes. The findings indicate (but sometimes merely suggest) that the measures are complementary and were associated with aspects of mental health. The authors also provided practical recommendations of how these findings could be implemented in the college sport setting to benefit athletes. It is unfortunate that the authors fail to cite the rich historical and empirical basis for the role of mental health athletes and its impact athletic performance which would buttress the general principle behind the study. This is important because for many years the field of sport psychology discounted the role of mental health in athletes. Relatedly, the rationale for the measures the authors ended up focusing on could be sharpened. Major concerns regarding this include the fact there has been an continues to be a lack of consensus as to 1) exactly what the construct of Mental Toughness is, 2) whether it is unidimensional or consists of multiple factors and, 3) whether sport-specific measures are more appropriate than broader (i.e., measures designed for all athletes or the general population). The result is that there are at least six major MT measures that range from one to 12 factors. There is also little similarity of the factors used in these various measures. This means that the specific MT findings in this paper may be particular to the unidimensional measure used and that other multi-factor measures could have yielded different results. While I am not suggesting the authors either used the wrong measures or should have employed multiple MT measures, they should at least acknowledge the present state of affairs in MT research as well as the possibility that a different measure of MT could have yielded different results.

Specific Comments

95 Why focusing on these two constructs at the exclusion of other relevant and validated measures that also have been found to be associated with positive mental health benefits? Have these measures been neglected? Have they been found to be more useful in related research? Has no one used them in tandem as you did in your study? More context would be helpful. You provided some information that supports the link of these measures but make no attempt at all to provide either theoretical or empirical evidence that these measures are more central to general mental health than other constructs. You also fail to acknowledge the theoretical basis of several of your hypotheses. For, example, hypothesis three in which you predict that D-I athletes will score higher in your mental health measures could (should) be traced back to the mental health model long ago posited by Morgan. This context would help deepen the theoretical basis for your study.

98 Are there in fact studies, theories, or perspectives that have been expressed in the literature suggesting or contending that MT and SC are competing, incongruent or orthogonal?

146 It might be pointed out that Loehr’s measure of MT suffers from serious validity issues, a problem not uncommon with sport-specific inventories.

150-156 This is the crux of the problem with MT, aside from the decidedly generic definitions of MT used in the literature, no one who studies it agrees on what precise the construct is. The debate goes beyond whether MT is unidimensional or has several factors. In fact, Gucciardi’s initial MT scale (the AFMTI Gucciardi, 2009) was composed of four sub-factors. Among researchers who continue to conceptualize MT as multifactorial, there is a lack of consensus as to both the number of sub-factors (they range from two to 12 among the six most commonly used measures of MT) or what those factors are. what those factors are or even the number of factors. Other researchers contend that MT measures should be specific to the target population; hence the development of sport-specific MT questionnaires. There is also the contention that MT shares considerable overlap with resilience, a construct of considerable currency in sport psychology research. None of this is to say that you either selected the wrong construct or wrong measure of a construct to employ in your study. But you have neglected to point out the issues surrounding the construct of MT fully as the perspectives voiced by critics of MT research. As stated by Jones (2002): “Mental toughness is probably one of the most used but least understood terms used in sport psychology.”

157 You need to provide citations that support the contention that Gucciardi’s more recent definition of MT has been gaining acceptance.

163-166 Are these correlations consistent across all MT measures? Have sex differences been consistently found across all MT measures? I would guess not. Also, the fact that the evidence you cite that MT scores are higher in male athletes is ‘preliminary’ implies that 1) MT measures do not have published norms and 2) early studies did not find sex differences.

175 Is this a consistent finding? Have others replicated this? Are the findings based on a particular MT measure? Is this the only study that has found a link with a negative outcome in athletes?

245-247 Based on points made earlier, you should elaborate on your rationale for choosing this scale to the exclusion of the six or so others widely used in sport research. Moreover there is an argument that sport-specific scales, in some cases, are not necessarily more efficacious than general measures that have proven construct validity and have been shown to be effective in a wide range of situations and populations (e.g., anxiety measures). Provide enough evidence that your reader doesn’t second guess your decision.

428-430 This statement seems like wishful thinking: the interaction is “buffering” yet not “statistically significant” but it still “warrants further elaboration”. It is perfectly acceptable to suggest that further research is needed, but you shouldn’t imply you found something in the absence of statistical significance.

432. “SC seems to facilitate the effective use of MT” You need to be clear here as to whether this is based on statistically significant findings or not.

434-437 This sounds an awful lot like a model of resiliency; the idea that it either increases or decreases (granted, the model also allows for no change) after experiencing adversity.

Reviewer #2: I thank you for the opportunity to review this manuscript. It addresses an important topic that too many athletes and coaches fail to understand, self-compassion. Relating self-compassion to mental toughness expands the literature in this area and the authors provides a recommendation connected to mindfulness. However, I think more specific recommendations for how to do this are warranted, particularly from the point of view of a sport psychology professional.

1. Throughout the paper the use of the term Mental Health (MH) is not consistent. For example, on Pg 4, Line 83, What is a MH environment? This is not the language used by the NCAA and is not accurate.

2. Pg 7 Line 165—167. How is MT interrelated to each of those concepts? Depression and thriving? This needs more explanation.

3. Why was it hypothesize that DI athletes would score higher in MT, SC, and MH? There doesn’t seem to be any prior evidence to support this hypothesis.

4. Much of the discussion focuses on how the results found that mental toughness is related to the mindfulness subscale of SC (.45), but there is no mention of the relationship to self-kindness (.42) which is nearly as strong. What explanation do the authors have for this relationship?

5. Was there an opportunity for participants to identify as a gender other than male or female? If not, how might that have affected the validity of the study?

6. Pg 20, line 432-433. “…SC seems to facilitate the effective use MT.” This doesn’t make sense. There is no such thing as an effective (or ineffective) use of mental toughness. Overall, some of the conclusions drawn about the relationship between MT and SC are too broad considering this study only found a moderate correlation between the two constructs. Some of these implications should be dialed back.

7. Pg 23, Line 483-489. This section would benefit from discussion of evolving gender norms. There are now many strong, female athlete role models that defy these norms, as well as more outspoken male athletes who have shared their struggles with mental health (e.g., Michael Phelps).

8. Pg 25, Line 531-533. “These recommendations improve best practices by providing specific and practical psychological steps for athletes, coaches, and stakeholders invested in the collective task of more fully realizing best practice No. 4.” The recommendations are not actually very specific. It would be helpful to provide more information on how to enhance MT and SC within student athletes.

9. I’m curious why there is not mention of sport psychologists or sport psychology consultants as they are the ones most likely to implement MT, SC, and/or mindfulness training?

6. PLOS authors have the option to publish the peer review history of their article (what does this mean?). If published, this will include your full peer review and any attached files.

Reviewer #1: No

Reviewer #2: No

---

## [Author Response · Author response to Decision Letter 0]

30 Nov 2020

Dear all-

We attached a new cover letter.

Thank you for all your help and support!

AS

---

## [Editor Report · Decision Letter 1]

14 Dec 2020

Can athletes be tough yet compassionate to themselves? Practical Implications for NCAA Mental Health Best Practice No. 4.

PONE-D-20-08466R1

Dear Dr. Stamatis,

We’re pleased to inform you that your manuscript has been judged scientifically suitable for publication and will be formally accepted for publication once it meets all outstanding technical requirements.

Kind regards,

Professor Dominic Micklewright, PhD CPsychol PFHEA FBASES FACSM

Academic Editor

PLOS ONE

---

## [Editor Report · Acceptance letter]

21 Dec 2020

PONE-D-20-08466R1 

Can athletes be tough yet compassionate to themselves? Practical Implications for NCAA Mental Health Best Practice No. 4. 

Dear Dr. Stamatis:

I'm pleased to inform you that your manuscript has been deemed suitable for publication in PLOS ONE. Congratulations! Your manuscript is now with our production department. 

Kind regards, 

on behalf of

Professor Dominic Micklewright 

Academic Editor

PLOS ONE